# Look Before You Leap: Safe Model-Based Reinforcement Learning with Human Intervention

**Yunkun Xu**[1,2], **Zhenyu Liu**[1]*, **Guifang Duan**[1], **Jiangcheng Zhu**[2], **Xiaolong Bai**[2], **Jianrong Tan**[1]
[1]State Key Laboratory of CAD&CG, Zhejiang University
[2]Huawei Cloud
{xuyunkun, liuzy, gfduan, egi}@zju.edu.cn, {zhujiangcheng, baixiaolong1}@huawei.com

**Abstract:** Safety has become one of the main challenges of applying deep reinforcement learning to real world systems. Currently, the incorporation of external knowledge such as human oversight is the only means to prevent the agent from visiting the catastrophic state. In this paper, we propose MBHI, a novel framework for safe model-based reinforcement learning, which ensures safety in the state-level and can effectively avoid both "local" and "non-local" catastrophes. An ensemble of supervised learners are trained in MBHI to imitate human blocking decisions. Similar to human decision-making process, MBHI will roll out an imagined trajectory in the dynamics model before executing actions to the environment, and estimate its safety. When the imagination encounters a catastrophe, MBHI will block the current action and use an efficient MPC method to output a safety policy. We evaluate our method on several safety tasks, and the results show that MBHI achieved better performance in terms of sample efficiency and number of catastrophes compared to the baselines.

**Keywords:** Safety RL, Model-based RL, Model Predict Control

## 1 Introduction

Deep reinforcement learning (DRL) is proposed as an automated framework for intelligent decision-making problems, and has shown tremendous progress in various domains such as video games [1, 2], board games [3, 4] and robotic control [5, 6]. However, most of these success were achieved in the idealized simulator where agents can interact with the environment without restriction. In real-world applications, random exploration may lead to equipment damage or even worse results [7]. Consequently, when applied in the real world, DRL algorithms need to ensure the security throughout the whole learning process, especially in safety-critical scenarios.

This has encouraged the research on safety in DRL, which aims to maximize the cumulative rewards while minimizing safety violations during the training processes [8, 9]. Safe RL can be divided into two categories: trajectory-based [9, 10, 11] and state-based [12, 13, 14]. The former uses the cost function to evaluate the safety of the whole trajectory, while state-based methods place restrictions on a state-wise basis. Safe RL algorithms such as CPO [10] and HIRL [12] have shown that it is possible to significantly reduce the number of constraint violations in safety-critical tasks.

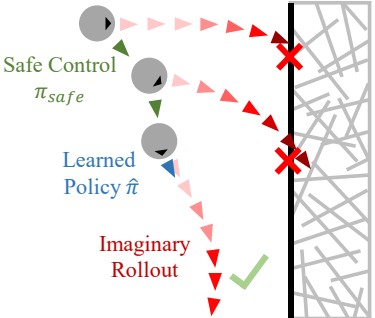

Figure 1: Imaginary safety detection and replacement of catastrophic actions in MBHI.

Unfortunately, most of existing methods have high sample complexity, which means that a large number of interactions with the real world are required, resulting in the increase of the number of catastrophe. Another shortcoming is that most of existing approaches use the explicitly defined constraint or safety cost function, while safety is subjective and always vague. For example, in autonomous driving, the overseer brakes when he judges that the car is going to crash. The overseer makes judgement based on subjective feelings rather than vehicle dynamic model. Finally, it is more strict and natural

---

*Corresponding author

5th Conference on Robot Learning (CoRL 2021), London, UK.

to define safety over the state. However, most of existing state-based methods are hard to effectively avoid potential disasters in the future [12, 13, 15], or need suboptimal demonstrations [14, 16].

To address the above-described challenges, in this paper, we introduce a novel safety-aware model-based framework, called Safe Model-based Reinforcement Learning with Human Intervention (MBHI), which incorporates safe RL with model-based methods and human knowledge. Intuitively, before excuting an action, people will estimate potential dangers in the future. In MBHI, an ensemble of deep neural networks is used to model a supervised Blocker to imitate the human overseer or explicit constraints, output the catastrophe probability of the state and evaluate the model uncertainty. As shown in Figure 1, at each time step, we roll out a trajectory with a certain number of steps in the learned dynamics to check the safety of the agent in the future. If dangerous, the MPC controller will take over and output a safe action to avoid possible future catastrophes in advance. Furthermore, safe

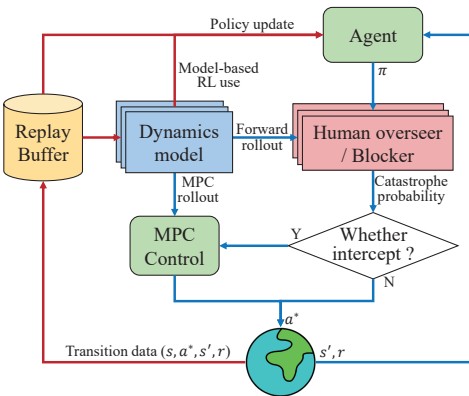

Figure 2: Overall framework of MBHI. The interaction phase is shown in blue and the policy learning in red.

active learning is integrated to enable grounded exploration only in safe regions, thereby accelerating model training on the premise of safety.

To summarize, the primary contributions of our work are as follows: (1) a novel safe model-based RL framework with human intervention, (2) safe active learning mechanism that only encourages agents to explore in safe regions, (3) a state-based action correction mechanism that perceive possible catastrophes in the future and avoid them in advance, (4) efficient MPC solving method that outputs a safe alternative policy in continuous action space.

## 2  Related Work

**Safe RL:** An interaction process is called safe if the agent rarely accesses the dangerous state [9, 11, 13]. Introducing external knowledge is the only way to ensure safety [12, 17]. Recently, various kinds of explicit constraints are used to impose some form of safety measures. To name some, constraints on expectation [9, 10], constraints on uncertainty [7, 18], and Conditional Value-at-Risk [11, 19, 20]. Unlike the above trajectory-based safety guarantees, [13] and [21] solve an optimization problem in the policy-level to ensure the safety of states. CARL [22] first trains the agent in non-safety-critical environments, and then adapts to the target environment. In the setting of implicit constraints, HIRL [12] and [15] train a supervised learner to imitate the human's intervention decisions and use it to evaluate the safety of the current state-action pairs. DDPGfD [16] and SAVED [14] learn the policy efficiently and safely in the sparse reward environment by using suboptimal human demonstrations. This work also focuses on learning implicit safety constraints from human knowledge. The main difference of MBHI is that (1) we image in the leaned dynamics to avoid potential hazards in the future, (2) the human knowledge only contains the judgment about catastrophes, not suboptimal demonstrations, and (3) non-safety-critical source environments are not available.

**Model-based RL:** Model-based RL exhibits low sample complexity and fast convergence. The learned dynamics model can be used for planning [4, 23], value expansion [24, 25], or imaginary training [26, 27, 28]. In planning approaches, MBMF [29] and PETS [23] use Model Predictive Control (MPC) to select actions based on the learned model. MVE [25] and STEVE [24] incorporate value expansion to improve the targets for temporal difference learning. ME-TRPO [26] and MBPO [27] are directly trained on imaginary data to accelerate policy learning. To limit the influence of the model error, the uncertainty estimation is widely used in MBRL. For instance, PETS [23] use uncertainty-aware models for trajectory propagation and planning. STEVE [24] interpolates between different imaginary rollout lengths to favor those targets with lower uncertainty. M2AC [30] uses the masking mechanism to only use the model when the model error is small.

In this work, STEVE is used as the base learning algorithm, but our framework is generally applicable to other model-based algorithms with uncertainty quantification, such as MBPO, M2AC, etc. We choose STEVE for three main reasons. Firstly, it explicitly estimates uncertainty for both dynamics and Q-function. Secondly, if the dynamics model cannot be learned or the Q-value network converges better, STEVE can reduce to model-free algorithm instead of learning from inaccurate imaginary data. Thirdly, since STEVE is an off-policy algorithm, the safety detection process in MBHI can be integrated with value expansion at each time-step to reduce calculation amount.

## 3 Preliminaries

### 3.1 Markov Decision Processes

We focus on Markov Decision Process (MDP) with continuous states and action spaces, defined as $(\mathcal{S}, \mathcal{A}, R, p_0, T, \gamma)$, in which $\mathcal{S}$ is the state space, $\mathcal{A}$ is the action space, $R : \mathcal{S} \times \mathcal{A} \times \mathcal{S} \to \mathbb{R}$ is the reward function, $p_0 : \mathcal{S} \to \mathbb{R}_+$ is the initial state distribution, $T : \mathcal{S} \times \mathcal{A} \to \mathcal{S}$ is the dynamics function, and $\gamma \in [0, 1]$ is the discount factor. The agent starts from an initial state $s_0 \sim p_0$, at each time step, the agent chooses an action $a_t \in \mathcal{A}$ in state $s_t \in \mathcal{S}$ according to a policy $\pi_\theta(s_t)$ parameterized with $\theta$, transitions to a successor state $s_{t+1} = T(s_t, a_t)$ and receives a reward $r_t = R(s_t, a_t, s_{t+1})$. The goal of reinforcement learning is to find an optimal policy $\pi_{\theta^*}$ that maximize the expected discounted sum of rewards $\theta^* = \arg\max_\theta \mathbb{E}_{s_0 \sim p_0}[\sum_{t=0}^{\infty} \gamma^t r_t]$.

### 3.2 Stochastic Ensemble Value Expansion

In model-based RL, the agent learns the world model from the replay buffer to approximate the true dynamics function. In STEVE, the dynamics model consists of the following modules:

$$
\begin{aligned}
\text{Transition model}: \quad & s_t \sim \hat{T}_\xi(s_t | s_{t-1}, a_{t-1}) \\
\text{Reward model}: \quad & r_t \sim \hat{R}_\phi(r_t | s_t, a_t, s_{t+1}) \\
\text{Termination model}: \quad & d_t \sim \hat{D}_\psi(d_t | s_t, a_t, s_{t+1}).
\end{aligned} \tag{1}
$$

An ensemble of parameters is maintained to estimate uncertainty in the learned dynamics (i.e. $\xi = \{\xi_i\}_{i=1}^{N_T}$, $\phi = \{\phi_i\}_{i=1}^{N_R}$, $\psi = \{\psi_i\}_{i=1}^{N_D}$). The entire dynamics model is optimized as follows:

$$
\min_{\xi, \phi, \psi} \mathbb{E}[||s_{t+1} - \hat{T}_\xi(s_t, a_t)||^2 + ||r_t - \hat{R}_\phi(s_t, a_t, s_{t+1})||^2 + \mathbb{H}(d_t, \hat{D}_\psi(s_t, a_t, s_{t+1})))], \tag{2}
$$

where $\mathbb{H}$ is the cross-entropy, and the expectation is calculated over data $\{(s_t, a_t, s_{t+1}, r_t, d_t)\}$.

For actor-critic methods or Q-learning methods, action-value function $\hat{Q}_\varphi$ is a critical quantity to guide policy updating. The Temporal-Difference target is written as:

$$
T^{TD}(r_t, s_{t+1}) = r_t + \gamma(1 - d_t)\hat{Q}_{\varphi^-}(s_{t+1}, \pi_\theta(s_{t+1})), \tag{3}
$$

where $\varphi^-$ means target network. STEVE improves TD targets by combining the short-term value estimate using the dynamics and long-term value estimate using $\hat{Q}_\varphi$. An ensemble of Q-functions is also maintained to evaluate the relative uncertainty between the dynamics and Q-function. Through these uncertainty estimates, STEVE dynamically interpolates between imagined rollouts of different horizon lengths, favoring those candidate targets with lower uncertainty, and give up utilizing them when significant errors are introduced. Thus, the interpolated target $T_H^{STEVE}$ is written as

$$
T_H^{STEVE}(r_t, s_{t+1}) = \sum_{h=0}^{H} T_h^\mu \cdot (T_h^{\sigma^2})^{-1} / \sum_j (T_j^{\sigma^2})^{-1} \tag{4}
$$

$$
T_h(r_t, s_{t+1}) = r_t + \sum_{i=1}^{h} D_i \gamma^i \hat{R}_\phi(s_{t+i}, a_{t+i}, s_{t+i+1}) + D_{h+1}\gamma^{h+1}\hat{Q}_{\varphi^-}(s_{t+h+1}, a_{t+h+1}), \tag{5}
$$

where $D_i = d_t \cdot \prod_{j=1}^{i}(1 - \hat{D}_\psi(s_{t+j}, a_{t+j}, s_{t+j+1}))$, $s_{t+h+1} = \hat{T}_\xi(s_{t+h}, a_{t+h})$. $T_h$ is the value expansion target with length $h$, $T_h^\mu$ and $T_h^{\sigma^2}$ are the empirical mean and variance for each $T_h$. If the environment is too complex or noise, STEVE will ignore the dynamics and reduce to model-free algorithm immediately.

# 4 Methods

In this section, we present a novel model-based safe RL framework, MBHI, as shown in Figure 2. In the interaction phase, at each time-step, we use the current policy network $\pi_\theta$ to roll out multiple steps in the learned dynamics, and check the safety of the imaginary trajectories with the supervised Blocker. If the catastrophe probability of the imaginary future rollout is greater than the threshold, the output action will be intercepted and replaced by the MPC controller. In policy learning phase, the model-based method is used to update the policy. Moreover, to overcome catastrophic forgetting problem of the network [12], we split replay buffer into safe and unsafe categories.

## 4.1 Catastrophe Prediction Network

Similar to HIRL [12], a supervised learner is trained to imitate the human overseer and block actions that are unsafe, but we maintain an ensemble of parameters for the Blocker $\mu = \{\mu_i\}_{i=1}^{N_B}$ to evaluate the uncertainty. In concrete terms, the human-imitator method has the following advantages. 1) Blocker can learn from data logs, human intervention data or explicit safety constraints by supervised learning. 2) Blocker is modular, it can be applied in distributed system and different tasks. 3) As binary network, the output value of the Blocker can be regarded as the distribution distance between the input state and the catastrophe zone. 4) Blocker can provide gradient information.

During human oversight phase, we store $(s_t, a_t, s_{t+1}, r_t, d_t, c_t)$ at each time-step, where $c_t$ is a binary label for whether the current state is dangerous defined by humans. Thus, the Blocker can be trained in a supervised manner by minimizing the cross entropy loss as:

$$\mathcal{L}_{\mu_i} = \mathbb{H}(c_t, \hat{B}_{\mu_i}(s_t, a_t, s_{t+1})) \qquad 1 \leqq i \leqq N_B. \tag{6}$$

Each $\hat{B}_{\mu_i}$ is initialized with various weights and trained on different input sequences. At this stage, for model-based RL, the collected data can also be used to train the dynamics using Eq (2). For the sake of fairness, in all experiments, we did not pre-train the dynamics model during the human oversight phase.

## 4.2 Safe Active Exploration

In order to reduce model uncertainty, we aim to query those points with large entropy, or where the disagreement would be most reduced in a posterior model. In our settings, the dynamics model takes the form of ensemble, and the disagreement can be quantified by calculating the empirical variance across the output of each model. Therefore, the optimization objective can be written as:

$$\pi_\theta = \max[\mathbb{E}_{\tau \sim \pi_\theta}[\sum_{t=0}^{T}((1-\lambda)r_t + \lambda c_a \sigma^2(\hat{T}_\xi(s_t, a_t)))]], \tag{7}$$

where $c_a$ is the coefficient of the active learning reward and $\lambda$ is a policy dependent weighting. The second term of Eq 7 can also be regarded as maximizing the differential entropy of the transition model. The model outputs is in the form of empirical Gaussian distribution, its differential entropy can be easily obtained, given by $H(x) = \frac{1}{2}(\ln(2\pi\sigma^2) + 1)$. Therefore, maximizing variance is equivalent to maximizing model entropy. It is also possible to maximize the variance of the predicted reward, as proposed in [31], but we find it perform poorly in the sparse reward environment.

However, for safe RL, we do not want agents to visit unsafe regions, even if these areas have greater uncertainty. A simple yet effective method is to mask the exploration reward near the dangerous area. The catastrophe probability predicted by the Blocker can be regarded as the distribution distance to the unsafe region. Thus, Eq (7) can be rewritten in the form of safety-aware.

$$\pi_\theta = \max[\mathbb{E}_{\tau \sim \pi_\theta}[\sum_{t=0}^{T}((1-\lambda)r_t + \lambda c_a \sigma^2(\hat{T}_\xi(s_t, a_t))(1 - \mathbb{E}_\mu[\hat{B}_\mu(s_t, a_t, s_{t+1})])^\alpha)]], \tag{8}$$

where the exponent $\alpha$ controls the masking rate. $\lambda$ determines the level of exploration guided by the learned transition model, it should tend towards stability with the convergence of the policy and the full exploration of the environment. In this work, we use the uncertainty measure of the dynamics to adjust $\lambda$ adaptively. Formally, $\lambda$ is defined as $\lambda_{t+1} = \frac{\bar{\sigma}_t^2(\hat{T}_\xi(\tau))}{2 \cdot \max_k \bar{\sigma}_{k<t}^2(\hat{T}_\xi(\tau))}$, where $\bar{\sigma}_t^2$ denotes the mean variance. This restricts $\lambda$ to $[0, 0.5]$ and make the exploration to be guided by the reward signal.

## 4.3 Replacement of Catastrophic Actions

Before the agent execute the action to the environment, a $C$ step trajectory from current state is rolled out in each dynamics. The catastrophe probability of the imagination is computed as follows.

$$p = \max_{c \in C} \mathbb{E}_{i \sim N_B, j \sim N_T} [\hat{B}_{\mu_i}(s^j_{c-1}, \pi_\theta(s^j_{c-1}), \hat{T}_{\xi_j}(s^j_{c-1}, \pi_\theta(s^j_{c-1})))]. \tag{9}$$

If $p$ is greater than the safety threshold, it is considered that the agent will have a disaster in a certain number of steps under the current policy, and it is necessary to replace the action in advance. In HIRL [12], the author selects the alternative action based on a lookup table or the logit score ranking. However, these methods are not suitable for continuous action space and "non-local" catastrophes. In this work, Model-Predictive Control is used as an expert to output safe actions. Given a state $s_t$, the MPC prediction horizon $H$ and the action sequence $a_{t:t+H-1}$, each dynamics model is used to predict the resulting trajectories $\{(s_{t+i}, r_{t+i-1})\}_{i=1}^{H}$. At each timestep $t$, the MPC controller solves the following optimization problem

$$\max_{a_t, \dots, a_{t+H-1}} \sum_{h=0}^{H-1} \hat{r}_{MPC}(s_{t+h}, a_{t+h}, s_{t+h+1}) \qquad s.t. \quad s_{t+h+1} = \hat{T}_\xi(s_{t+h}, a_{t+h}), \tag{10}$$

where $\hat{r}_{MPC}$ is the MPC reward. There are two requirements of it. First, guide the agent complete the obstacle-independent task. Second, guide the agent away from the unsafe regions. In consequence, $\hat{r}_{MPC}$ contains the predicted external reward and the predicted disaster probability. In addition, we further introduce a Leave-One-Out safety estimation to make MPC samples as far away from the unsafe region as possible. Given the unsafe dataset $\mathcal{D}_{us}$ and the MPC sampled state $s_{t+h}$, the estimator is defined as $u_{t+h} = D_{KL}[p(\cdot|\mathcal{D}_{us} \cup s_{t+h})||p(\cdot|\mathcal{D}_{us})]$. The value of $u_{t+h}$ is close to zero, when $p(\cdot|\mathcal{D}_{us} \cup s_{t+h})$ and $p(\cdot|\mathcal{D}_{us})$ gradually match each other, which indicates that the sampled state is dangerous. Therefore, we aim to maximize $\sum_{h=1}^{H} u_{t+h}$ to keep the whole MPC sampled sequence away from danger. The MPC return can be formulated as

$$\hat{r}_{MPC}(s_{t+h}, a_{t+h}, s_{t+h+1}) = \mathbb{E}_\phi[\hat{R}_\phi(s_{t+h}, a_{t+h}, s_{t+h+1})] + \mathbb{E}_\mu[\hat{B}_\mu(s_{t+h}, a_{t+h}, s_{t+h+1})] + u_{t+h+1}. \tag{11}$$

Finally, the controller executes the first action $a_t$ to the environment, advances to the next time-step, and recheck the safety of the agent.

To improve the efficiency and performance of MPC, especially in high-dimensional action space, as shown in Figure 3, the gradient of the dynamics is used to guide the update of CEM sampling distribution. Each iteration consists of three steps: (1) Sample action sequences from CEM sampling distribution, and calculate MPC return with Eq (11). (2) Fix the dynamics parameters, and update action sequences via gradient ascent method. (3) Update the parameters of CEM sampling distribution. The action replacement mechanism is summarized in algorithm 1 in Appendix A.

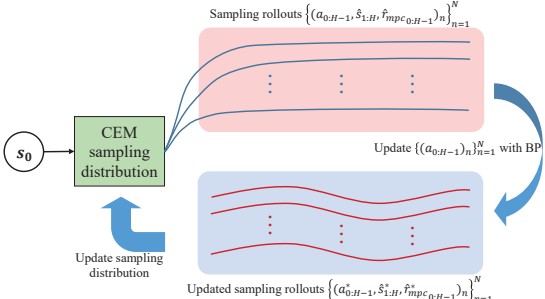

Figure 3: Schematic of MPC optimization method.

A negative reward must be given to guide the agent to learn a safe policy. The interception mechanism in MBHI considers the danger of the current policy in state $s_t$ rather than the state-action pair. Thus, the negative reward should solely depend on the state. In this work, an intrinsic reward signal is generated based on how far the agent is from the unsafe region to encourage the agent to stay away from these areas. Specifically, the output of Blocker is served as this kind of measurement. Besides, to increase the punishment near the unsafe area, a boundary value $Bound$ is introduced. When the predicted catastrophe probability is greater than this value, a larger scaling factor is used to increase the negative reward. The intrinsic reward signal is computed as

$$r^i(s_t, a_t, s_{t+1}) = \begin{cases} -c_l \mathbb{E}_\mu[\hat{B}_\mu(s_t, a_t, s_{t+1})] & if \ \mathbb{E}_\mu[\hat{B}_\mu(s_t, a_t, s_{t+1})] < Bound \\ -c_h \mathbb{E}_\mu[\hat{B}_\mu(s_t, a_t, s_{t+1})] & else, \end{cases} \tag{12}$$

where $c_l$ and $c_h$ are scaling factors, which control the penalties in different areas. $Bound$ is the safety bound, a smaller bound will increase the complexity of the environment and obtain a more

conservative policy. Finally, the overall optimization problem can be written as

$$\theta^* = \arg\max_\theta \mathbb{E}_{a_t \sim \pi_\theta(s_t)}[\sum_{t=0}^{\infty} \gamma^t((1-\lambda)(r_t + r^i(s_t, a_t, s_{t+1})) + \lambda r^a(s_t, a_t, s_{t+1}))], \qquad (13)$$

where $r^a$ is the active learning reward in Eq (8). The full procedure is outlined in Algorithm 2 in Appendix A.

## 5  Experiments

### 5.1  Experiment Setting

In this section, we evaluate our approach and various baselines over five safety-critical continuous control tasks. 1) PuckWorld-L & PuckWorld-H: they differ in acceleration. The catastrophe of PuckWorld-H is locally avoidable, while PuckWorld-L is not. The agent is rewarded for reaching the target and is constrained to not touch the barrier. 2) Reacher: the agent is rewarded for reaching the target without touching the vertical bar. 3) Ant-Limit: the agent needs to move forward between two parallel barriers. 4) Ant-Block: the agent needs to move forward and avoid obstacles in the path. The last three tasks are adapted from PyBullet's environments [32]. The visualization and details of these environments are given in Appendix B. All experiments are evaluated over 5 random seeds.

We compare our proposed MBHI against both the model-based and model-free baselines, PPO, DDPG and STEVE, and human intervention method, HIRL [12]. we extend HIRL to the continuous action space and also use STEVE as the basic algorithm. To ensure a fair comparison, HIRL and MBHI have the same Blocker and action replacement mechanism. HIRL also splits the replay buffer like MBHI. The final implementation differences between HIRL and MBHI are summarized as follows: (1) HIRL only receives a negative reward when a catastrophic action is blocked, while penalty is introduced in MBHI in the form of Eq (12); (2) HIRL does not employ active exploration; (3) HIRL does not image in the learned dynamics to detect potential disasters in the future. In all experiments, we use the same network structure of the policy, value, dynamics and Blocker for all algorithms, the number of ensemble models is 4. Details can be found in Appendix H.

### 5.2  Comparison of Performance

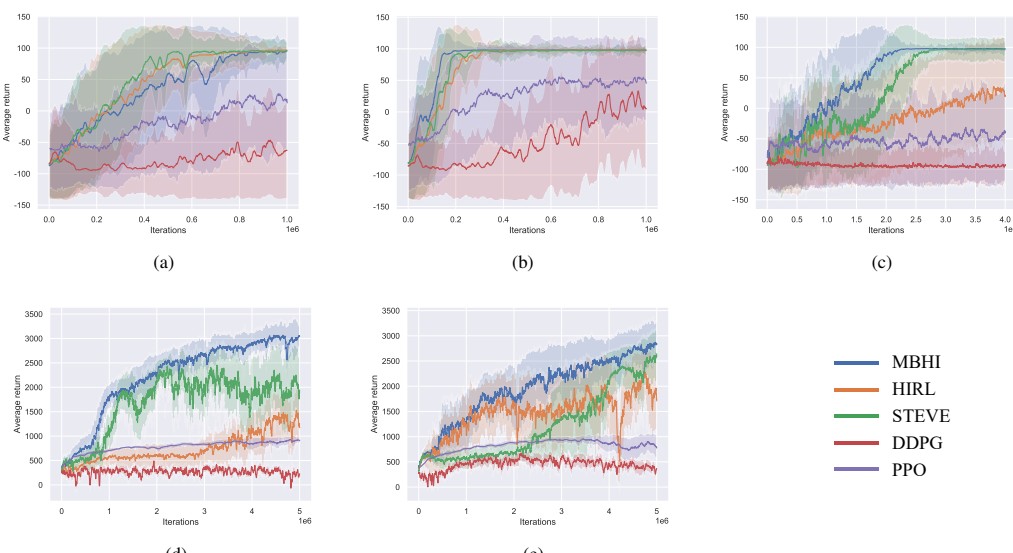

Figure 4: Learning curves of MBHI and baselines on five continuous control tasks over time (mean and standard deviation). (a) PuckWorld-L, (b) PuckWorld-H, (c) Reacher, (d) Ant-Limit, (e) Ant-Block. The x-axis is environment steps.

The comparison results in Fig 4 demonstrate that the sample efficiency of MBHI is better than baselines, especially DDPG and PPO. MBHI achieves similar or better performances in all tasks

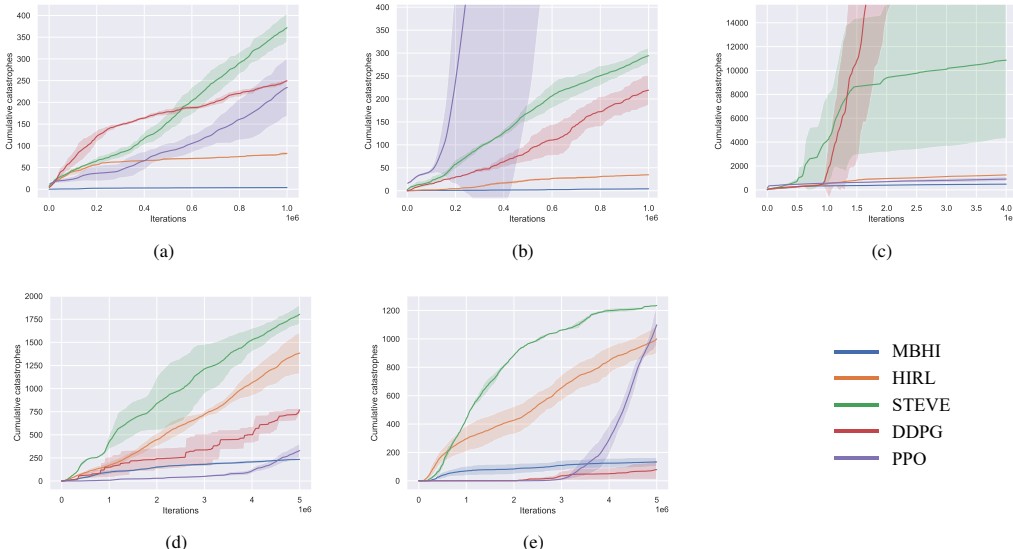

Figure 5: Cumulative catastrophes of MBHI and baselines on five continuous control tasks over time (mean and standard deviation). (a) PuckWorld-L, (b) PuckWorld-H, (c) Reacher, (d) Ant-Limit, (e) Ant-Block. The x-axis is environment steps.

than baselines. In PuckWorld-H, the parameters of MBHI and HIRL in safety inspection are exactly the same, but MBHI can converge faster, which illustrates the importance of active exploration. Frequent action shielding increases the complexity of policy learning. However, as an off-policy method, we do not find perceptible negative effects for MBHI.

The experiment results of STEVE and DDPG in Fig 5 demonstrate that optimal unsupervised behaviors can result in a large number of catastrophes. Note that DDPG and PPO has fewer catastrophes in some environments because the agent has not learned how to complete the task, and tends to move near the initial state. For example, in Ant-Block, the agent does not even successfully move to the place where it could encounter the obstacle, which can explain why the number of catastrophes of PPO suddenly rises in Fig 5e. Additionally, HIRL is able to successfully resolve locally avoidable catastrophes, but failed in other environments. By contrast, MBHI effectively avoids both "local" and "non-local" catastrophes during training. In the simple environment like PuckWorld, MBHI almost achieves zero catastrophes.

In tasks other than PuckWorld-H, HIRL cannot effectively save the agent. However, from Figure 5, we can see that the number of catastrophes in HIRL is significantly less than STEVE. This is because we divide the replay buffer in HIRL into safe and unsafe parts. Besides, we can also find that even though STEVE has converged, the number of catastrophes is still rising. Our findings are in line with recent results on catastrophic forgetting [12, 33]. MBHI can well restrain this problem.

### 5.3 Ablation on Safety Detection Rollout Length

Intuitively, the longer the safety detection distance, the more conservative the policy. We evaluate different safety detection rollout lengths on PuckWorld-L and PuckWorld-H. The result is shown in Figure 6. It shows that, surprisingly, when the safety detection distance is sufficient, our method is not so sensitive to the length of safety detection rollouts. This also proves that safety detection in the imagination can effectively avoid the catastrophe that will happen in advance. In addition, the long detection distance does not further reduce the cumulative catastrophes significantly, but it will introduce unnecessary calculation costs.

### 5.4 Comparison of Action Replacement Methods

We study the impact of different MPC optimization techniques and reward settings. The compared optimizers include the uniform random search [29] and the cross-entropy method [23]. The MPC

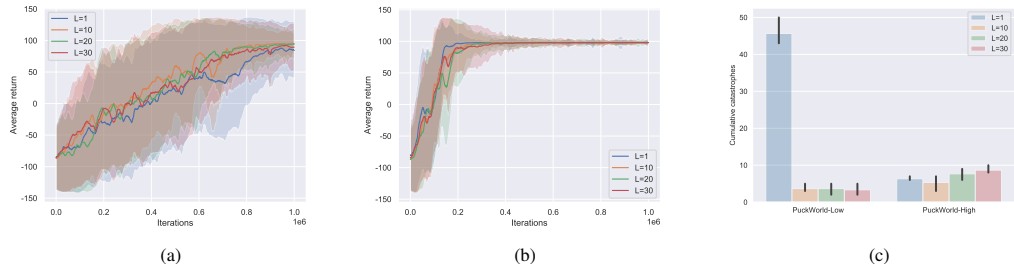

(a)                                    (b)                                    (c)

Figure 6: Ablation studies of the safety detection rollout length. (a) Average return in PuckWorld-L; (b) Average return in PuckWorld-H; (c) Cumulative catastrophes. Error bars are 95% bootstrap confidence intervals.

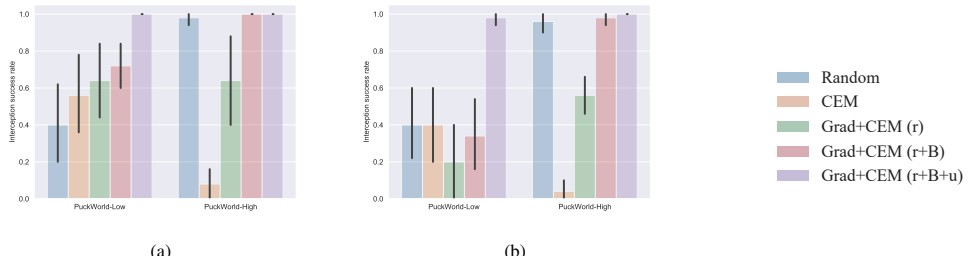

(a)                                                        (b)

Figure 7: Interception success rate of different methods. (a) Converged dynamics model, (b) pre-trained dynamics model. For fair comparison, random shooting sampled 3200 times, CEM has 25 iterations of sampling 128 candidate actions, and Grad+CEM has 5 iterations plus 5 gradient steps of sampling 128 candidate actions. Shown is the mean performance, with error bars showing 95% bootstrap confidence intervals over 5 runs.

reward settings include three types: 1) the reward $\hat{r}$ predicted by the learned reward function; 2) $\hat{r}$ and the catastrophe probability $\hat{B}$ predicted by the Blocker; 3) $\hat{r}$, $\hat{B}$, and the distribution distance $u$ between MPC samples and unsafe samples. For the sake of fairness, we also use $\hat{r} + \hat{B} + u$ as the MPC reward in Random and CEM. We sample ten points uniformly around the catastrophe as the initial position, and make the agent move toward the catastrophe at the fastest speed. In addition, the converged dynamics model and the pre-trained model are used as the simulator of MPC to evaluate the robustness of the proposed method. The results of the interception success rate are shown in Figure 7. It shows that our method has better interception performance and is more robust.

## 6 Conclusion

In this paper, we proposed a state-based safety oversight mechanism MBHI, which is based on the model-based RL algorithm with active learning method. MBHI is forward-looking, it considers the safety of the current policy in a certain time-step in the future, replaces dangerous actions in advance, and realizes the prevention of both "local" and "non-local" catastrophes. We evaluated the proposed algorithm on five continuous safety control tasks, the experiment results demonstrate that MBHI can significantly reduce catastrophes and accelerate training. Furthermore, our solution acts directly at the policy level, so it is independent of the learning algorithm and can be plugged into any existing model-based and model-free continuous control algorithms.

MBHI effectively alleviates the problem of "non-local" catastrophes, but it needs to have a certain understanding of the task. The safety detection distance needs to be long enough to avoid catastrophes in advance. Besides, hyperparameters are sensitive to the performance of the Blocker. Going forward, the safety detection length should be adaptive to the specific task, rather than set a fixed value. Moreover, in future research, we plan to study the theoretical guarantee of MBHI and improve the algorithm's robustness to the hyper-parameters. In terms of computation, since safety evaluation inevitably increases the cost of training process, we plan to study more effective methods to estimate the potential danger of the agent in the future and to integrate these estimates in model-based policy optimization.

**Acknowledgments**

We would like to thank Yang Li and our anonymous reviewers for their insightful comments and suggestions. This work was supported in part by the National Key Research and Development Program of China (grant No. 2019YFB1312600), in part by the National Natural Science Foundation of China (grant No. 52075480), in part by the High-level Talent Special Support Plan of Zhejiang Province, China (grant No. 2020R52004) and in part by the Natural Science Foundation of Zhejiang Province, China under Grant Y19E050078.

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
