# OpenReview forum: "Look Before You Leap: Safe Model-Based Reinforcement Learning with Human Intervention"
_robot-learning.org/CoRL/2021/Conference — CoRL2021 Poster_

### Official Review · Reviewer_wEtd · 2021-07-04

**Originality:** Very Good
**Technical Quality:** Very Good
**Clarity Of Presentation:** Good
**Impact:** 3

**Recommendation:**

Strong Accept: I recommend accepting the paper and will argue for my recommendation even if other reviewers hold a different opinion.

**Summary:**

The paper proposes a method which utilizes a policy guided by dynamics and value-function predictions and uncertainties over unsafe states for safer exploration and learning.


**Issues:**

See weaknesses/minor issues section.

**Reviewer Expertise:**

Good: General knowledge of the area

**Strengths And Weaknesses:**

**Paper Strengths**
Technical contribution: The method is an amalgamation of different ideas that results in a well-performing algorithm which demonstrates improvements over a number of baselines. While no part itself is particularly novel, the combination of everything makes the method a novel contribution.

Experiments: Experimental evaluation was solid with respect to the baselines and environments tested. The method performs well in terms of catastrophe avoidance, and offers mixed to strong improvements over baselines on final reward.

Paper Writing: Given the complexity of the method, the method is generally described well.

**Paper Weaknesses**
Missing related work: Cautious Adaptation for RL in Safety-Critical Settings (Zhang et al 2020) also targets state-based safety through an ensemble of dynamics models with MPC planning, and SAVED (Thananjeyan et al 2020) performs safe RL from human demonstrations, with a similar value-function and dynamics-model based cost function. These are important recent works the authors should cite and contrast their method with in the Related Works/intro section.

Claim: “However, existing state-based approaches only consider the catastrophe of locally avoidable, that is, catastrophes can be avoided by choosing the appropriate action just before the catastrophe would have happened.” I don’t think this can be stated as fact. Model-based methods with MPC planning will be able to avoid catastrophes over longer horizons as long as the horizon of the planner is long enough. This is even more true for Dynamics + Value-function based const functions which exist in both STEVE and the aforementioned SAVED work.

Clarity: $T_{STEVE}^H$ seems to come out of nowhere (Eq 5) and its purpose is not explained immediately, and $T_h^\mu$ also should be mentioned.

Ablations: The authors introduce a method that has many different design choices, such as the leave-one-out estimation, gradient updates for CEM, having c_l and c_h, the safety bound, etc. There should be more ablation studies and sensitivity analysis on these parameters in general.

Experiments: I can’t find where the paper mentions the number of random seeds the results are reported over. And as hinted to in the related works weaknesses, there are baselines that can be extended to work on the scenario in the paper that would be nice to compare against.


**Minor Issues**
Typo: Safe Active Explorationm (4.2), Puck-Workd (L214),


**Summary Of Recommendation:**

I recommend acceptance of this paper due to the solid performance and the technical contribution of the method. However, I do have some issues listed in the paper weaknesses section that I would like to be addressed during the rebuttal.

---

> ### Author Response · Authors · 2021-08-30
> **Reply to Reviewer wEtd - Part 1**
>
> 1. Missing related work: Cautious Adaptation for RL in Safety-Critical Settings (Zhang et al 2020) also targets state-based safety through an ensemble of dynamics models with MPC planning, and SAVED (Thananjeyan et al 2020) performs safe RL from human demonstrations, with a similar value-function and dynamics-model based cost function. These are important recent works the authors should cite and contrast their method with in the Related Works/intro section.
>
>     We have cited and contrasted these works in the "Related works". Compared with CARL, the main difference of MBHI is that we do not need to pre-train in non-safety-critical environments, which is also difficult to obtain in real scenarios. Compared with SAVED, the main difference is that the human knowledge required by MBHI only contains the human judgment about catastrophes, not human demonstrations. The suboptimal and safety trajectories guided by the human are not available. In MBHI, the training data provided by humans is not the complete and suboptimal trajectory. In many cases, we think it is also difficult for humans to provide feasible complete and suboptimal demonstrations.
>
> 2. Claim: "However, existing state-based approaches only consider the catastrophe of locally avoidable, that is, catastrophes can be avoided by choosing the appropriate action just before the catastrophe would have happened." I don’t think this can be stated as fact. Model-based methods with MPC planning will be able to avoid catastrophes over longer horizons as long as the horizon of the planner is long enough. This is even more true for Dynamics + Value-function based const functions which exist in both STEVE and the aforementioned SAVED work.
>
> 	We have revised this sentence to make the description more accurately. The learned dynamics in MBRL methods must visit dangerous areas in large numbers to model catastrophe, which is not allowed in safety-critical environment. MPC planning is an effective method, and it is also used in this work to correct dangerous actions. However, it is inefficient if we always use MPC to output an action. Besides, MPC can hardly deal with the sparse reward environment. In SAVED, human demonstration is introduced to guide the MPC controller to overcome the problem of sparse cost feedback, but this is not available in MBHI. Dynamics + value-function is more efficient and flexible than dynamics + MPC, but has a higher tolerance for disasters. The experimental results in Figure 5 show that STEVE has large number of catastrophes in all experiments.
>
> 3. Clarity: $T_H^{STEVE}$ seems to come out of nowhere (Eq 5) and its purpose is not explained immediately, and $T_h^\mu$ also should be mentioned.
>
>     We have revised the last paragraph of Section 3.2 to describe Equations 4 and 5 more clearly.
>
> 4. Ablations: The authors introduce a method that has many different design choices, such as the leave-one-out estimation, gradient updates for CEM, having c_l and c_h, the safety bound, etc. There should be more ablation studies and sensitivity analysis on these parameters in general.
>
> 	The ablation studies of leave-one-out estimation and Grad + CEM have been done in Section 5.4 and presented in Figure 7. Different MPC reward settings are studied with Grad + CEM sampling method, the MPC reward with leave-one-out estimation is more robust and performs better. In the comparison experiment of Random Shooting, CEM and Grad + CEM, for fairness, all sampling methods used the full MPC reward (i.e. $r+B+u$), and the performance of Grad + CEM is significantly better. Furthermore, we have added ablation studies of scaling factors ($c_l$ and $c_h$) and safety bound, the visualization of parameter lambda, the comparison of interception points and labeled unsafe area, and the analysis of the key components of MBHI in Appendix D, E, F, G.
>
> 5. Experiments: I can’t find where the paper mentions the number of random seeds the results are reported over. And as hinted to in the related works weaknesses, there are baselines that can be extended to work on the scenario in the paper that would be nice to compare against.
>
>     We have added a description of the number of random seeds at the end of the first paragraph of Section 5.1. As explained before, the main advantage of MBHI is that it can learn from human subjective judgments, which is difficult to be formulated explicitly. Besides, the human knowledge required by MBHI only contains the judgment about catastrophes, not the advance human knowledge like suboptimal demonstrations. Moreover, the need of non-safety-critical source environments in CARL is also different from our hypothesis. Therefore, we only compare with the baseline based on human imitator. It is interesting to compare with the baseline with policy constraints, and we have added the comparison of PPO in Section 5.1 and 5.2.
>
> 6. Typo: Safe Active Explorationm (4.2), Puck-Workd (L214),
>
>    Thanks again, and we have corrected these typos in the manuscript.

---

> > ### Comment · Reviewer_wEtd · 2021-09-03
> > **Response**
> >
> > Thanks for the reply. I believe most of my complaints have been addressed and have updated my score appropriately.
> > I believe this paper is solid and should be accepted.

---

> ### Author Response · Authors · 2021-08-30
> **Reply to Reviewer wEtd - Part 2**
>
> For a more intuitive presentation, we have listed the main changes in the manuscript.
>
> Main paper:
> ```
> (1) fixed typos and polished up the text.
>
> (2) uniformed the notations in the equations, and made all notations explained.
>
> (3) cited and contrasted more related works including "Leveraging Demonstrations for Deep Reinforcement Learning on Robotics Problems with Sparse Rewards" (Mel Vecerik et al. 2018), "Cautious Adaptation For Reinforcement Learning in Safety-Critical Settings" (Jesse Zhang et al. 2020), and "Safety Augmented Value Estimation from Demonstrations (SAVED): Safe Deep Model-Based RL for Sparse Cost Robotic Tasks" (Brijen Thananjeyan et al. 2020) in "Related Works".
>
> (4) added more baseline details and the description of random seeds in Section 5.1.
>
> (5) added comparison result of PPO in Section 5.1 and 5.2.
> ```
>
> Supplement:
> ```
> (1) added details of the unsafe areas in each task in Appendix B.
>
> (2) added workload analysis of human overseer in Appendix C.
>
> (3) added visualization of the active learning parameter \lambda in Appendix D, Figure 9.
>
> (4) added ablation studies of scaling factors (c_l and c_h) in Appendix E.1.
>
> (5) added ablation studies of the safety bound in Appendix E.2.
>
> (6) added discussions of the key components of MBHI in Appendix F.
>
> (7) added visualization of labeled areas, interception points and the predicted catastrophe probability in Appendix G, Figure 12.
>
> (8) added more implementation details in Appendix H.
> ```

---

### Official Review · Reviewer_VQ9u · 2021-07-21

**Originality:** Good
**Technical Quality:** Good
**Clarity Of Presentation:** Good
**Impact:** 3

**Recommendation:**

Weak Accept: I recommend accepting the paper, but will not argue for my recommendation if the majority of other reviewers have a different opinion.

**Summary:**

This work deals with the safe RL problem.  The authors propose a method where a model of the environment is learned and used to simulate fixed-length rollouts from a current state.  With these rollouts, a supervised learning framework is derived where it is predicted if the rollout is dangerous (based on human-defined labels).  If the rollout is dangerous, MPC control is used to avoid the dangerous state (select safe actions).  The work is highly relevant to robot learning and is validated on the standard suite of MuJoCo safe RL environments.  Compared to other models such as HIRL, this work deals with longer simulated trajectories rather than a single transition so it can reason about non-local danger.

**Issues:**

See above

**Reviewer Expertise:**

Good: General knowledge of the area

**Strengths And Weaknesses:**

This method is well validated on highly relevant tasks where safety is an important aspect and results are promising.  This method performs well in terms of avoiding catastrophes, but performance in terms of return is not highly significant in most tasks compared with a baseline model-based approach (especially in the non-local catastrophe environment).  There is an absence of real robotic simulations, but I do not think this diminishes the quality of the work. The ablation experiments are useful and highlight the specific impact of the method, though more could be added on parameter sensitivity.

The work appears to be sound in terms of the method. The improvements are simple/incremental over HIRL but the work is shown to be empirically useful for the problem it attacks, avoiding non-local danger.  It is validated on the non-local danger tasks explicitly.  The novelty of the method overall is lacking, but this work combines many different models to form a useful and relevant method.

More comparisons against traditional safe RL algorithms could be present, such as a policy optimization method with conditional value at risk objective (CVaR). Comparing against CVaR would make a useful comparison against a method not requiring human intervention, which is difficult to obtain.  I would also have liked to see some comparisons against methods with more constrained updates such as TRPO or PPO as opposed to DDPG.

The text is generally polished.  There are a few minor details to point out:
- The authors alternate between the notation s’ and s_{t+x} between the method and related work section.
- \alpha in (9) is superscript and appears to be a power instead of a weight.
- In (14) tau is written instead of the symbol.
- The parameters for each part of the dynamics model could be combined into a single set with its own symbol?
- Notation could be simplified at many steps such as (13) where the bound is not stated.
- It is unclear how many random seeds are used.

Overall, the visuals are very useful in understanding the method. The paper reads clearly.


**Summary Of Recommendation:**

I think this work is borderline.  The method is relevant to CoRL but is lacking some key evaluation baselines and novelty in general. The authors should be able to make minor adjustments to the text and overall, the paper is relatively easy to understand.

**Post Response**:  While I still think this work is borderline, due to the revisions from the authors I am now leaning towards acceptance.  I believe the additional baseline and ablation analysis solidify the usefulness of the method and believe the paper should be "accepted if there is room".

---

> ### Author Response · Authors · 2021-08-30
> **Reply to Reviewer VQ9u - Part 1**
>
> 1. More comparisons against traditional safe RL algorithms could be present, such as a policy optimization method with conditional value at risk objective (CVaR). Comparing against CVaR would make a useful comparison against a method not requiring human intervention, which is difficult to obtain. I would also have liked to see some comparisons against methods with more constrained updates such as TRPO or PPO as opposed to DDPG.
>
>     Unlike most existing constraint-based methods, the assumption of this work is that explicit safety constraints are not available. One of the main advantage of MBHI is that it can learn from human subjective judgments, which is difficult to be formulated explicitly. Compared with explicit constraints, the Blocker modeled using neural networks is more flexible but less accurate. It is interesting to compare with the baseline with policy constraints, and we have added the comparison of PPO in Section 5.1 and 5.2.
>
> 2. The text is generally polished. There are a few minor details to point out: The authors alternate between the notation s’ and s_{t+x} between the method and related work section. \alpha in (9) is superscript and appears to be a power instead of a weight. In (14) tau is written instead of the symbol. The parameters for each part of the dynamics model could be combined into a single set with its own symbol? Notation could be simplified at many steps such as (13) where the bound is not stated. It is unclear how many random seeds are used.
>
>     Thanks again for your careful review. We have revised the related formulas and descriptions in the manuscript, and added a description of the number of random seeds at the end of the first paragraph of Section 5.1.

---

> ### Author Response · Authors · 2021-08-30
> **Reply to Reviewer VQ9u - Part 2**
>
> For a more intuitive presentation, we have listed the main changes in the manuscript.
>
> Main paper:
> ```
> (1) fixed typos and polished up the text.
>
> (2) uniformed the notations in the equations, and made all notations explained.
>
> (3) cited and contrasted more related works including "Leveraging Demonstrations for Deep Reinforcement Learning on Robotics Problems with Sparse Rewards" (Mel Vecerik et al. 2018), "Cautious Adaptation For Reinforcement Learning in Safety-Critical Settings" (Jesse Zhang et al. 2020), and "Safety Augmented Value Estimation from Demonstrations (SAVED): Safe Deep Model-Based RL for Sparse Cost Robotic Tasks" (Brijen Thananjeyan et al. 2020) in "Related Works".
>
> (4) added more baseline details and the description of random seeds in Section 5.1.
>
> (5) added comparison result of PPO in Section 5.1 and 5.2.
> ```
>
> Supplement:
> ```
> (1) added details of the unsafe areas in each task in Appendix B.
>
> (2) added workload analysis of human overseer in Appendix C.
>
> (3) added visualization of the active learning parameter \lambda in Appendix D, Figure 9.
>
> (4) added ablation studies of scaling factors (c_l and c_h) in Appendix E.1.
>
> (5) added ablation studies of the safety bound in Appendix E.2.
>
> (6) added discussions of the key components of MBHI in Appendix F.
>
> (7) added visualization of labeled areas, interception points and the predicted catastrophe probability in Appendix G, Figure 12.
>
> (8) added more implementation details in Appendix H.
> ```

---

### Official Review · Reviewer_QSAz · 2021-07-22

**Originality:** Good
**Technical Quality:** Very Good
**Clarity Of Presentation:** Very Good
**Impact:** 4

**Recommendation:**

Weak Accept: I recommend accepting the paper, but will not argue for my recommendation if the majority of other reviewers have a different opinion.

**Summary:**

The paper presents a way to perform model-based RL with catastrophe action blocking and active correction. It builds on the MBRL algorithm STEVE (which comes with uncertainty quantification), and also human-in-the-loop method HIRL (which trains a human blocker model), with the additional technique of unrolling imaginary trajectories and optimize the future trajectory. The algorithm is able to outperform several baselines in catastrophe prevention and reward maximization.

**Issues:**

- Has the authors considered other types of human-in-the-loop algorithms besides HIRL for catastrophe detection? Would be curious for any other existing work in the field
- More details could be given in the Appendix / supplementary materials section on algorithm and experiment details.


**Reviewer Expertise:**

Fair: Some knowledge of the area

**Strengths And Weaknesses:**

I think the main strengths of the paper are:
- Novel technique of unrolling imagined trajectory
- Good literature review of the field
- Promising experimental results

The weaknesses:
- Experiment details: There could be more details on the algorithm and experiment settings. For example, it is unclear what's the human workload during the human supervision phase.
- Baseline comparison: Since this method involves human blocking, there could be other types of human-in-the-loop algorithms serving as baseline comparisons. For example, instead of naive DDPG, DDPGfD* might also serve as a fair comparison since that involves human demonstrations rather than just RL from scratch.

* Leveraging Demonstrations for Deep Reinforcement Learning on Robotics Problems with Sparse Rewards
Mel Vecerik, Todd Hester, Jonathan Scholz, Fumin Wang, Olivier Pietquin, Bilal Piot, Nicolas Heess, Thomas Rothörl, Thomas Lampe, Martin Riedmiller

**Summary Of Recommendation:**

I would recommend the paper because of its insight and novelty, the introduction of human supervision and blocking in MBRL and extensive experimental results. It could potentially benefit from more algorithmic details and baseline comparisons as mentioned above but overall I think it is an insightful work.

---

> ### Author Response · Authors · 2021-08-30
> **Reply to Reviewer QSAz - Part 1**
>
> Thank you very much for the useful and constructive comments. We have revised the manuscript and added more experiments, ablations and analysis. The modifications are detailed below
>
> 1. Experiment details: There could be more details on the algorithm and experiment settings. For example, it is unclear what's the human workload during the human supervision phase.
>
>     We have enriched the experiment settings and implementation details in Section 5.1, Appendix B and H. Moreover, we have described the approximate human workload during the human supervision phase in Appendix C.
>
> 2. Baseline comparison: Since this method involves human blocking, there could be other types of human-in-the-loop algorithms serving as baseline comparisons. For example, instead of naive DDPG, DDPGfD* might also serve as a fair comparison since that involves human demonstrations rather than just RL from scratch. ——Leveraging Demonstrations for Deep Reinforcement Learning on Robotics Problems with Sparse Rewards.
>
>     The human knowledge required by MBHI only contains the human judgment about catastrophes, instead of human demonstrations. The suboptimal and safety trajectories guided by the human are not available. In MBHI, the training data provided by humans is not the complete and suboptimal trajectory. In many cases, we think it is also difficult for humans to provide feasible complete and suboptimal demonstrations.
>
> 3. Has the authors considered other types of human-in-the-loop algorithms besides HIRL for catastrophe detection? Would be curious for any other existing work in the field
>
>     As far as we know, in safety reinforcement learning, human-in-the-loop methods are mainly divided into two categories. One is to obtain disaster judgments from humans, and the other is to obtain suboptimal demonstration. Both MBHI and HIRL introduce human knowledge about disasters by modeling a supervised learner to imitate the human overseer. We believe that the data on human intervention is easily accessible. Another way to introduce human knowledge is demonstration, which we have cited and contrasted in the "Related works". In demonstration method like SAVED (Thananjeyan et al 2020), suboptimal human demonstration is introduced to constrain exploration to regions in which the agent is confident in task completion, and provide signals about task progress in sparse reward environments. MPC with CEM sampling method is used to output safety actions under constraints.
>
> 4. More details could be given in the Appendix / supplementary materials section on algorithm and experiment details.
>
>     We have enriched the experiments and implementation details in Appendix B and H. Furthermore, We have also added the ablation experiments of scaling factors and safety bound, the visualization of the active learning parameter lambda, the comparison of interception points and labeled unsafe area, and the analysis of the key components of MBHI in Appendix D, E, F, G.

---

> ### Author Response · Authors · 2021-08-30
> **Reply to Reviewer QSAz - Part 2**
>
> For a more intuitive presentation, we have listed the main changes in the manuscript.
>
> Main paper:
> ```
> (1) fixed typos and polished up the text.
>
> (2) uniformed the notations in the equations, and made all notations explained.
>
> (3) cited and contrasted more related works including "Leveraging Demonstrations for Deep Reinforcement Learning on Robotics Problems with Sparse Rewards" (Mel Vecerik et al. 2018), "Cautious Adaptation For Reinforcement Learning in Safety-Critical Settings" (Jesse Zhang et al. 2020), and "Safety Augmented Value Estimation from Demonstrations (SAVED): Safe Deep Model-Based RL for Sparse Cost Robotic Tasks" (Brijen Thananjeyan et al. 2020) in "Related Works".
>
> (4) added more baseline details and the description of random seeds in Section 5.1.
>
> (5) added comparison result of PPO in Section 5.1 and 5.2.
> ```
>
> Supplement:
> ```
> (1) added details of the unsafe areas in each task in Appendix B.
>
> (2) added workload analysis of human overseer in Appendix C.
>
> (3) added visualization of the active learning parameter \lambda in Appendix D, Figure 9.
>
> (4) added ablation studies of scaling factors (c_l and c_h) in Appendix E.1.
>
> (5) added ablation studies of the safety bound in Appendix E.2.
>
> (6) added discussions of the key components of MBHI in Appendix F.
>
> (7) added visualization of labeled areas, interception points and the predicted catastrophe probability in Appendix G, Figure 12.
>
> (8) added more implementation details in Appendix H.
> ```

---

### Official Review · Reviewer_rwsy · 2021-07-31

**Originality:** Fair
**Technical Quality:** Very Good
**Clarity Of Presentation:** Good
**Impact:** 3

**Recommendation:**

Strong Accept: I recommend accepting the paper and will argue for my recommendation even if other reviewers hold a different opinion.

**Summary:**

This paper presents a method to avoid potential future catastrophes in a reinforcement learning setting.
It uses the framework STEVE because it explicitly estimates the uncertainty for the dynamics but any other MBRL with uncertainty quantification could be used.

The method works by checking for failures in the imaginary rollouts and then replacing the faulty actions with an MPC controller. The catastrophe prediction is done with an ensemble of networks trained by supervised learning with human labels, called the Blocker Network.
The loss for model learning consists of:
- the loss used in STEVE
- an additional cross entropy loss for their binary network which classifies whether the current transition is safe or not (supervised learning with human labels)

The expected reward optimized by the policy network consists of three terms:
- the standard reward from the environment
- an exploration term (model entropy) that is proportional to the probability of *not* being in a dangerous region
- a safe-learning reward that explicitly penalizes unsafe areas


**Issues:**

Regarding Figure 9 in the Appendix: what we see are the replaced actions or the original ones? What are the unsafe areas in these cases?


You don’t really show how accurate the blocker network is. It could be interesting to add a plot with the labeled areas and compare to the regions where actions are replaced. As a reader, it is not clear what the labels are. Are only the obstacles labeled as unsafe?

It is not clear to me if the obstacles are modeled by the dynamics.

Instead of using a blocker network, did you try explicitly adding a penalty cost as a distance to the obstacles?

What is the final difference between your modification of HIRL and MBHI? Is it the blocker network and the buffer split? This information should be made explicit.

Line 158, what do you mean by “asymptotically consistent”?
Line 162, did you try to plot lambda over time? It could be interesting to see.
Line 189, it is mostly clear what is the use of the KL divergence term, namely to sample away from unsafe regions, but it could be better explained.


Many typos, please correct.
e.g.:
line 43: the catastrophe of locally avoidable
or line 260: sampling samples
Line 263: and the model only be pre-trained


**Reviewer Expertise:**

Good: General knowledge of the area

**Strengths And Weaknesses:**

### Strengths

This method is able to avoid non-local catastrophes, meaning that the decision to avoid failure is done many steps ahead rather than right before the critical point.
The single components are not novel but their combination is novel and works better than the presented baselines.

### Weaknesses
The method consists of many components and it is not very easy to put everything in place, the presentation is a bit confusing and it could be improved.


I believe the authors could compare their work to other baselines that also employ active exploration.
By reading the paper it seems that this work is the first one to have both active exploration and safety awareness. I would rephrase the introduction/related work and add more comparisons to existing methods. The presented model-based ones are more an ablation of MBHI.
The authors could better summarize the necessary additions - that are now scattered in the paper - and present some take home messages. For example, if separating the buffer is an important addition (which is briefly mentioned in line 244).

Additional comments listed in Issues.


**Summary Of Recommendation:**

I believe the paper could be better organized and optimized for clarity. Some details are missing and should be included. Some baselines are reported in the results but nothing that specifically includes active exploration and risk awareness.
I think the paper has potential: the idea of avoiding future catastrophes in advance and to include human intervention is valid. I will modify my score upon improvement of the paper.

Edit: Updated to Strong Accept after rebuttal

---

> ### Author Response · Authors · 2021-08-30
> **Reply to Reviewer rwsy - Part 1**
>
> Thank you very much for the useful and insightful suggestions. We have revised the manuscript and added more experiments, ablations and analysis. The modifications are detailed below
>
> 1. The method consists of many components and it is not very easy to put everything in place, the presentation is a bit confusing and it could be improved.
>
>     We have summarized the main advantage of MBHI and added a discussion of the key components of MBHI in Appendix F. Including Look before leap, Separated replay buffer, Safety-aware intrinsic reward and Model-based RL. Besides, We have also added the ablation experiments of scaling factors and safety bound, the visualization of the active learning parameter lambda, and the comparison of interception points and labeled unsafe area in Appendix D, E, G.
>
> 2. I believe the authors could compare their work to other baselines that also employ active exploration.
>
>     In this work, like many existing MBRL, active learning method is introduced to encourage the agent to query points with large model entropy, so as to reduce model uncertainty, and ultimately to generate more accurate imaginary rollouts to improve learning efficiency. However, as discussed in SAMBA (A. Cowen-Rivers et al 2020), the desire for safety adds a complication, because a safety learning algorithm can hardly access unsafe areas. Therefore, there is greater uncertainty in unsafe areas, which will encourage agents to explore dangerous areas. This is not allowed in the safety-critical environment.
>
> 3. I would rephrase the introduction/related work and add more comparisons to existing methods.
>
>     We have enriched the "Introduction" and "Related Works" to cite and contrast other existing human-in-the-loop methods.
>
> 4. The presented model-based ones are more an ablation of MBHI. The authors could better summarize the necessary additions - that are now scattered in the paper - and present some take home messages. For example, if separating the buffer is an important addition (which is briefly mentioned in line 244).
>
>     As answered in Question 1, we have summarized the necessary components of MBHI in Appendix F.
>
> 5. Regarding Figure 9 in the Appendix: what we see are the replaced actions or the original ones? What are the unsafe areas in these cases?
>
>     We have added a clearer description to Figure 9. The dangerous actions in the trajectory shown in Figure 9 have been replaced by MPC controller. Moreover, We have also enriched Appendix B to describe in detail what catastrophes are in these environments. In PuckWorkd, it is a catastrophe if the puck enters the black area; in Reacher, it is a catastrophe if the arm collides with the vertical obstacle; in Ant-Limit, it is a catastrophe if the ant's centroid crosses the boundary; in Ant-Block, it is a catastrophe if the ant's centroid collides with the obstacle.
>
> 6. You don’t really show how accurate the blocker network is. It could be interesting to add a plot with the labeled areas and compare to the regions where actions are replaced. As a reader, it is not clear what the labels are. Are only the obstacles labeled as unsafe?
>
>     In this work, the catastrophe is not complex, and the Blocker can be trained easily. Besides, since the Blocker is an ensemble of fully-connected network, the slight difference in model accuracy has little effect on MBHI. In Appendix G, we have added several plots with interception points, labeled area and the prediction of the Blocker. In Appendix C, we have described which samples are labeled as unsafe. In this work, only the obstacle was labeled as unsafe, which greatly reduces the workload of human labors (no need to deduce) and can quickly evaluate the proposed method. Like the termination model in STEVE, since the input of the Blocker is $(s_t,a_t,s_{t+1})$ in practice, the transition can also be intercepted and labeled in advance. In this case, the labeled unsafe transition means that once it is visited, the agent cannot be rescued by humans.
>
> 7. It is not clear to me if the obstacles are modeled by the dynamics.
>
>    The dynamics consists of transition model, reward model and termination model. When catastrophes are frequently visited, the reward model and termination model can learn about obstacles, but this is not allowed in safety-critical environment. In this work, we additionally trained a model Blocker to imitate human interventions and predict whether the current transition will cause disaster, like a human overseer.

---

> > ### Comment · Reviewer_rwsy · 2021-09-03
> > **Response to authors**
> >
> > I believe the authors extensively addressed most of my concerns and extended the paper with relevant additions and comments. I update my recommendation to Strong Accept.

---

> ### Author Response · Authors · 2021-08-30
> **Reply to Reviewer rwsy - Part 2**
>
> 8. Instead of using a blocker network, did you try explicitly adding a penalty cost as a distance to the obstacles?
>
>    One of the main advantage of MBHI is that it can learn from human subjective judgments, which is difficult to be formulated explicitly. We didn't explicitly add a penalty to indicate the distance from the obstacles. In addition, as described in Sec 4.1, the Blocker can also be trained with explicit safety constraints, which is more efficient because it does not require human supervision. However, if we replace the Blocker with explicit constraints, it is difficult for the Blocker to guide CEM updates through backpropagation, which will affect the correction of dangerous actions.
>
> 9. What is the final difference between your modification of HIRL and MBHI? Is it the blocker network and the buffer split? This information should be made explicit.
>
>     We have detailed the difference of HIRL and MBHI in the last paragraph of Sec 5.1. For the sake of fairness, HIRL and MBHI have the same Blocker and action replacement mechanism. HIRL also splits the replay buffer like MBHI. The final implementation difference between HIRL and MBHI is summarized as follows: (1) HIRL only receives a negative reward when a catastrophic action is blocked, while penalty is introduced in MBHI in the form of Eq (12); (2) HIRL does not employ active exploration; (3) HIRL does not image in the learned dynamics to detect potential disasters in the future.
>
> 10. Line 158, what do you mean by "asymptotically consistent" ?
>
>     The phrase of this sentence is not accurate enough, we have revised this sentence to make the description clearer. With the convergence of the policy and the full exploration of the environment, the value of lambda should tend towards stability.
>
> 11. Line 162, did you try to plot lambda over time? It could be interesting to see.
>
> 	We have added the plot of lambda over time in all environments in Appendix D. In both PuckWorld and Reacher, $\lambda$ will gradually converge to zero. While in Ant, due to the particularity of the environment (unlimited space), $\lambda$ cannot converge to zero, but it will eventually stabilize at around 0.3.
>
> 12. Line 189, it is mostly clear what is the use of the KL divergence term, namely to sample away from unsafe regions, but it could be better explained.
>
>     We have enriched the explanation of KL divergence term.
>
> 13. Many typos, please correct. e.g.: line 43: the catastrophe of locally avoidable or line 260: sampling samples Line 263: and the model only be pre-trained.
>
>     Thanks again for your careful review, we have corrected these typos and grammatical errors in the manuscript.

---

> ### Author Response · Authors · 2021-08-30
> **Reply to Reviewer rwsy - Part 3**
>
> For a more intuitive presentation, we have listed the main changes in the manuscript.
>
> Main paper:
> ```
> (1) fixed typos and polished up the text.
>
> (2) uniformed the notations in the equations, and made all notations explained.
>
> (3) cited and contrasted more related works including "Leveraging Demonstrations for Deep Reinforcement Learning on Robotics Problems with Sparse Rewards" (Mel Vecerik et al. 2018), "Cautious Adaptation For Reinforcement Learning in Safety-Critical Settings" (Jesse Zhang et al. 2020), and "Safety Augmented Value Estimation from Demonstrations (SAVED): Safe Deep Model-Based RL for Sparse Cost Robotic Tasks" (Brijen Thananjeyan et al. 2020) in "Related Works".
>
> (4) added more baseline details and the description of random seeds in Section 5.1.
>
> (5) added comparison result of PPO in Section 5.1 and 5.2.
> ```
>
> Supplement:
> ```
> (1) added details of the unsafe areas in each task in Appendix B.
>
> (2) added workload analysis of human overseer in Appendix C.
>
> (3) added visualization of the active learning parameter \lambda in Appendix D, Figure 9.
>
> (4) added ablation studies of scaling factors (c_l and c_h) in Appendix E.1.
>
> (5) added ablation studies of the safety bound in Appendix E.2.
>
> (6) added discussions of the key components of MBHI in Appendix F.
>
> (7) added visualization of labeled areas, interception points and the predicted catastrophe probability in Appendix G, Figure 12.
>
> (8) added more implementation details in Appendix H.
> ```

---

### Meta-Review · Area_Chair_Qsp4 · 2021-08-10

**Recommendation:** Accept (Poster)
**Confidence:** 4

**Metareview:**

This paper proposes a safe reinforcement learning method, which switches to a safe MPC controller if unsafe events are predicted to happen. Reviewers generally agree that the paper makes good technical contributions to the important problem of safe learning. While most of the components of this algorithm are not novel, reviewers acknowledge that the combination of them all is novel and useful. The original concerns from reviewers about presentation, literature reviews, and baselines comparisons are well addressed in the rebuttal.

---

> ### Author Response · Authors · 2021-08-30
> **Reply to Area Chair Qsp4**
>
> Dear Area Chair.
>
> We sincerely appreciate the helpful comments and suggestions. We have added more experiments, ablation studies, visualizations and analysis in the manuscript. For a more intuitive presentation, we have listed the main changes in the manuscript.
>
> Main paper:
> ```
> (1) fixed typos and polished up the text.
>
> (2) uniformed the notations in the equations, and made all notations explained.
>
> (3) cited and contrasted more related works including "Leveraging Demonstrations for Deep Reinforcement Learning on Robotics Problems with Sparse Rewards" (Mel Vecerik et al. 2018), "Cautious Adaptation For Reinforcement Learning in Safety-Critical Settings" (Jesse Zhang et al. 2020), and "Safety Augmented Value Estimation from Demonstrations (SAVED): Safe Deep Model-Based RL for Sparse Cost Robotic Tasks" (Brijen Thananjeyan et al. 2020) in "Related Works".
>
> (4) added more baseline details and the description of random seeds in Section 5.1.
>
> (5) added comparison result of PPO in Section 5.1 and 5.2.
> ```
>
> Supplement:
> ```
> (1) added details of the unsafe areas in each task in Appendix B.
>
> (2) added workload analysis of human overseer in Appendix C.
>
> (3) added visualization of the active learning parameter \lambda in Appendix D, Figure 9.
>
> (4) added ablation studies of scaling factors (c_l and c_h) in Appendix E.1.
>
> (5) added ablation studies of the safety bound in Appendix E.2.
>
> (6) added discussions of the key components of MBHI in Appendix F.
>
> (7) added visualization of labeled areas, interception points and the predicted catastrophe probability in Appendix G, Figure 12.
>
> (8) added more implementation details in Appendix H.
> ```

---

### Decision · Program_Chairs · 2021-09-13

**Decision:**

Accept (Poster)

**Comment:**

This paper proposes a safe reinforcement learning method, which switches to a safe MPC controller if unsafe events are predicted to happen. Reviewers generally agree that the paper makes good technical contributions to the important problem of safe learning. While most of the components of this algorithm are not novel, reviewers acknowledge that the combination of them all is novel and useful. The original concerns from reviewers about presentation, literature reviews, and baselines comparisons are well addressed in the rebuttal.